# Pregnancy-Specific Beta-1-Glycoprotein 1 Increases HTR-8/SVneo Cell Migration through the Orai1/Akt Signaling Pathway

**DOI:** 10.3390/biom14030293

**Published:** 2024-02-29

**Authors:** Qunhua Wang, Yan Fang, Yuan Li, Huali Liu, Maoni Zhu, Xue Hu, Jinzhuo Zhou, Anqi Deng, Bing Shen, Hongbo Chen

**Affiliations:** 1Department of Obstetrics and Gynaecology, Maternal and Child Health Hospital Affiliated to Anhui Medical University, Hefei 230001, China; wangqunhua@ustc.edu.cn (Q.W.); 2145012060@stu.ahmu.edu.cn (Y.F.); liyuan@stu.ahmu.edu.cn (Y.L.); 2145012067@stu.ahmu.edu.cn (H.L.); 2245012234@stu.ahmu.edu.cn (M.Z.); 2345012360@stu.ahmu.edu.cn (X.H.); 2345012367@stu.ahmu.edu.cn (J.Z.); 2Department of Obstetrics and Gynaecology, The First Affiliated Hospital of USTC, Division of Life Sciences and Medicine, University of Science and Technology of China, Hefei 230031, China; 3The Fifth Clinical College of Anhui Medical University, Hefei 230032, China; 4School of Basic Medical Science, Anhui Medical University, Hefei 230032, China; 2113010546@stu.ahmu.edu.cn; 5Dr. Neher’s Biophysics Laboratory for Innovative Drug Discovery, State Key Laboratory of Quality Research in Chinese Medicine, Macau University of Science and Technology, Taipa, Macao SAR, China

**Keywords:** trophoblast, migration, PSG1, Orai1, Akt

## Abstract

The impaired invasion ability of trophoblast cells is related to the occurrence of preeclampsia (PE). We previously found that pregnancy-specific beta-1-glycoprotein 1 (PSG1) levels were decreased in the serum of individuals with early-onset preeclampsia (EOPE). This study investigated the effect of PSG1 on Orai1-mediated store-operated calcium entry (SOCE) and the Akt signaling pathway in human trophoblast cell migration. An enzyme-linked immunosorbent assay (ELISA) was used to determine the level of PSG1 in the serum of pregnant women with EOPE. The effects of PSG1 on trophoblast proliferation and migration were examined using cell counting kit-8 (CCK8) and wound healing experiments, respectively. The expression levels of Orai1, Akt, and phosphorylated Akt (p-Akt) were determined through Western blotting. The results confirmed that the serum PSG1 levels were lower in EOPE women than in healthy pregnant women. The PSG1 treatment upregulated the protein expression of Orai1 and p-Akt. The selective inhibitor of Orai1 (MRS1845) weakened the migration-promoting effect mediated by PSG1 via suppressing the Akt signaling pathway. Our findings revealed one of the mechanisms possibly involved in EOPE pathophysiology, which was that downregulated PSG1 may reduce the Orai1/Akt signaling pathway, thereby inhibiting trophoblast migration. PSG1 may serve as a potential target for the treatment and diagnosis of EOPE.

## 1. Introduction

Preeclampsia (PE) is a pregnancy-specific illness that can seriously harm both the mother and the fetus. It is characterized by new-onset hypertension and proteinuria, which appear after 20 weeks of pregnancy and impact several organ systems [1]. PE affects 2% to 8% of pregnancies worldwide and causes significant maternal and perinatal morbidity and mortality [2,3] PE has two subtypes: early-onset preeclampsia (EOPE) and late-onset preeclampsia (LOPE). Although EOPE and LOPE share similar clinical symptoms, these two subtypes of PE lead to different outcomes. EOPE, although less common, is associated with higher rates of neonatal mortality and a greater degree of maternal morbidity compared to LOPE [4]. Therefore, EOPE has received more attention from clinical researchers. Although the exact cause is not yet clear, it is widely believed that the insufficient invasion of trophoblasts is one of the important pathological factors in PE. In PE, the migration and invasion abilities of trophoblast cells are reduced, and remodeling of the uterine spiral arteries is impaired, leading to placental ischemia and hypoxia, causing a series of oxidative stress reactions, thereby causing a series of several symptoms of PE [5,6,7]. A study has shown that impaired placental development in early pregnancy and subsequent growth restriction are often associated with EOPE, while LOPE is likely associated with maternal endothelial dysfunction [8]. Therefore, elucidating the molecular mechanisms underlying the reduced migration ability of trophoblast cells is of great value in terms of PE therapy.

Pregnancy-specific glycoproteins (PSGs) share strong sequence similarity with the carcinoembryonic antigen (CEA) family characteristic of pregnancy and are mainly secreted by trophoblast cells [9]. PSGs are the most abundant placental-derived proteins in maternal blood during human pregnancy [10]. Among them, it has been demonstrated that PSG1 is a specific protein synthesized by placental syncytiotrophoblasts and secreted into the maternal blood, playing a potential role in placental vascular morphogenesis [11]. We previously found a decrease in the serum level of PSG1 in patients with EOPE [12]. In addition, abnormal placental development is an important pathological mechanism of EOPE. This finding raises the possibility that PSG1 may be involved in EOPE pathophysiology. However, the functional mechanism of PSG1 in the trophoblast of preeclampsia is still unclear.

Store-operated Ca^2+^ entry (SOCE) is the primary pathway for Ca^2+^ entry into almost all animal cells [13]. Many cellular activities have been revealed to be regulated by Ca^2+^ via SOCE in previous research [14,15]. For a long time, the increase in intracellular Ca^2+^ concentration ([Ca^2+^]_i_) caused by Ca^2+^ influx has been considered a key factor in phenotypic changes in cells, and changes in Orai expression are believed to regulate migration in different cells [16]. Orai1 belongs to the Orai family and is an important component of store-operated channels. Studies have demonstrated that Orai1 is involved in a number of physiological and pathological processes, such as proliferation and cell migration [17,18,19]. However, there are few studies on the transmission of Ca^2+^ signals in human trophoblasts. The literature has also shown that Akt can be activated by Orai1-mediated Ca^2+^ signaling, and Akt-related signaling pathways are associated with the pathogenesis of preeclampsia [20,21]. The imbalance of the Akt signaling pathway is related to abnormal placental development [22]. The migration of trophoblasts may be impacted by the downregulation of phosphorylated Akt (p-Akt) in extravillous trophoblasts [23]. Therefore, these findings imply that the Akt signaling pathway, which is also implicated in trophoblast migration, may be involved in the pathophysiology of EOPE. At present, the contribution of Orai1-mediated Ca^2+^ signaling, which activates Akt and modifies trophoblast cell activity, to the onset and progression of PE has not been documented.

The purpose of this study is to explore the hypothesis that PSG1 contributes to the pathophysiology of EOPE and the trophoblast migratory mechanism via the Orai1/Akt pathway. The expression of PSG1 in the serum of pregnant women was measured using the enzyme-linked immunosorbent assay (ELISA), and the results were compared between the serum levels of healthy pregnant women and those of women who had EOPE. Furthermore, we investigated how PSG1 affected the migration and proliferation of human trophoblast cells and observed that the Orai1/Akt pathway was involved in these processes.

## 2. Materials and Methods

### 2.1. Study Subjects

Ten age-matched normal pregnant women and seven women with EOPE were enlisted from the Maternal and Child Health Hospital Affiliated to Anhui Medical University (Anhui Medical University, Hefei, China). All study subjects underwent prenatal examinations between April 2022 and April 2023 and did not have any additional conditions such as infection, diabetes during pregnancy, or multiple pregnancies that could have an impact on their serum protein levels. The protocol of this study was implemented in accordance with the regulations of the Ethics Committee of Anhui Maternal and Child Health Hospital (20211221-ahky-001). The diagnosis of EOPE followed the International Federation of Gynecology and Obstetrics (FIGO) guidelines in 2019 [24].

### 2.2. ELISA Assay

A 2 mL blood sample was taken the morning after a pregnant patient was admitted without medicine, and it was kept in the dark at a room temperature (22–25 °C) setting for 1 h. After that, the supernatant was centrifuged for 15 min at 3000 rpm in a 4 °C centrifuge. Using a pipette, the supernatant was moved to a fresh test tube and kept at −80 °C. According to the instructions of the ELISA kit (#1202222836, Ray Biotech, Inc., Guangzhou China), the PSG1 concentration was measured in samples of EDTA plasma. The serum samples of the EOPE group and the normal control group were quantitatively analyzed according to the experimental steps. The minimum dose at which the kit can detect human PSG1 was 28.2 pg/mL.

### 2.3. Cell Culture

Human trophoblast cells, or HTR-8/SVneo cells, were acquired from Shanghai, China Enzyme-linked Biotechnology Co., Ltd., with permission from Professor Charles H. Graham from Queen’s University, Kingston, ON, Canada. Cells were cultivated at 37 °C in a 5% CO_2_ cell incubator with RIPM1640 media that included 10% fetal bovine serum and 1% penicillin–streptomycin. The culture medium was changed every 2 days, and when the cells reached 60–80% confluency, they were digested and passaged with a pancreatic enzyme. For the experiments, HTR-8/SVneo cells were cultured in media containing different concentrations of the recombinant human PSG1 protein (#9334-P1-050, R&D Systems, Minneapolis, MN, USA) for 24 h.

### 2.4. Wound Healing Assay

HTR-8/SVneo cells were seeded at an appropriate density in a plate and cultivated until 90–100% confluence. Scratch “wounds” were created using a 200 μL pipette tip. Next, the original culture medium was discarded, and the cells were cultured in a serum-free basic culture medium. In some experiments, the cells were treated with MRS1845 or ZSTK474 with or without PSG1 for 24 h. Scratch wound healing was then observed, and photos were taken after 0 and 24 h of the same place along the scratched edges.

### 2.5. Cell Viability Assay

Cell proliferation was assessed using the Cell Counting Kit-8 (CCK8) (#C0005, TOPSCIENCE, Shanghai, China). HTR-8/SVneo cells were cultured in a 96-well plate at 5000 cells/well with a RIPM1640 medium supplying 10% fetal bovine serum. After the cells adhered to the wall, corresponding treatment factors were added and incubated in the incubator for 24 h. Each well received 10 μL of CCK8 reagent, which was then incubated for 120 min. An automated enzyme-labeling apparatus (Rayto, Shenzhen, China) was used to detect the absorbance at a wavelength of 450 nm in order to determine the optical density (OD) value of the associated group.

### 2.6. Ca^2+^ Imaging

HTR-8/SVneo cells were grown on glass coverslips until they reached 50–60% confluence on 12-well plates. In some experiments, the cells were treated with PSG1 (0.1 μg/mL) and/or MRS1845 (10 μmol/L) for 24 h. To load the Ca^2+^-selective fluorescent indicator, the cells were incubated with 6 μmol/L of Fluo-8 AM for 30 min at 37 °C prior to the experiment. After this, the cells were placed in a buffer solution without Ca^2+^ (OPSS, 140 mM of NaCl, 5 mM of KCl, 1 mM of MgCl_2_, 10 mM of glucose, and 5 mM of HEPES; pH 7.3 to 7.4, adjusted with NaOH). The cells were treated with 4 μmol/L of thapsigargin (TG) for 10 min to deplete Ca^2+^ storage in the endoplasmic reticulum, followed by the addition of 2 mmol/L of CaCl_2_ to induce extracellular Ca^2+^ influx. Fluorescence was measured and viewed using a Nikon T200 fluorescence microscope (Tokyo, Japan). Fluo-8’s excitation and emission wavelengths were 488 nm and 515 nm, respectively. Before extracellular Ca^2+^ was applied, the baseline was designated as F0. Changes in [Ca^2+^]_i_ are expressed as the ratio of fluorescence to the intensity at baseline, or F_1_/F_0_.

### 2.7. Transcriptome Sequencing and Data Analysis

The experiment used three biological repeats and treated HTR-8/SVneo cells with PSG1 (0.1 μg/mL) for 24 h. Quality testing was conducted on the samples to ensure that they met the standards required for sequencing. Total RNA was isolated and purified using Trizol reagent (Invitrogen, Carlsbad, CA, USA) following the manufacturer’s procedure. The RNA amount and purity of each sample were quantified using NanoDrop ND-1000 (NanoDrop, Wilmington, DE, USA). The RNA integrity was assessed with an RIN number >7.0 and confirmed via electrophoresis with denaturing agarose gel electrophoresis Oligo (dT) magnetic beads were used to purify mRNA, and then the purified mRNA was sheared using a fragmentation buffer. The fragmented mRNAs were then used for the first-strand cDNA synthesis using reverse transcriptase and random hexamer primers. The second-strand cDNA was synthesized using DNA polymerase I and RNase H. After the fragments were ligated to adaptors, they were isolated as polymerase chain reaction (PCR) templates. Ultimately, a cDNA library was established with an average insert size of 300 ± 50 bp. The Clean Data were compared to the genome for analysis. StringTie software (stringtie-1.3.4d. Linux_x86_64) was used to perform the initial assembly of genes or transcripts and merge the initial assembly results of all samples. The transcripts were compared with reference annotations to obtain the final assembly annotation results. A significant difference analysis was performed between samples using the R package edgeR, defining genes with difference multiples (FC) > 2 times or FC < 0.5 times and a *p*-value < 0.05 as differentially expressed genes (DEGs) and performing GO and KEGG enrichment analyses on them.

### 2.8. RNA Preparation and Quantitative Real-Time PCR

Total RNA was extracted from cells using Trizol in accordance with the manufacturer’s instructions. cDNA was synthesized using the SPARKScript II RT Plus kit (AG0304-B, SparkJade, Qingdao, China) in accordance with the manufacturer’s instructions. The mRNA level of Orai1 was detected using a 2 × SYBR Green qPCR kit (AH0104-B, SparkJade, China). Roche Light Cycler 96 real-time PCR equipment was used to conduct the RT-qPCR test in accordance with the manual. GAPDH was used as an internal control. The primers were as follows: Orai1 (F: 5′-GGACGCTGACCGACTAC-3′, R: 5′-GGGACTCCTTGACCGAGTT-3′) and GAPDH (F: 5′-GGGCGAGATCCTCCAAAAT-3′, reverse: 5′-GGCTGTGTCATACTTTCATGG-3′). This measurement was conducted independently in triplicate.

### 2.9. Western Blotting

HTR-8/SVneo cells were collected and lysed using RIPA buffer (#P0013B; Beyotime Biotech, Haimen, China) to extract total proteins. In some experiments, the cells were treated with MRS1845 or ZSTK474 with or without PSG1 for 24 h. Following their separation via 10% SDS-PAGE electrophoresis, proteins were transferred onto polyvinylidene fluoride membranes. After being blocked with 5% non-fat milk for 2 h, the membranes were incubated overnight at 4 °C with primary antibodies. The antibodies included rabbit anti-Orai1 (1:1000, #13130-1-AP, Proteintech, Wuhan, China), rabbit anti-β-tubulin (1:5000, #P07437, Affinity Biosciences, Liyang, China), rabbit anti-Akt (1:1000, #T55561, Abmart, Shanghai, China), and rabbit anti-p-Akt (1:1000, #T40067, Abmart, China). For 2 h, the membrane was incubated with an HRP-labeled goat anti-rabbit IgG (H + L) secondary antibody (1:5000, #PR30009, Proteintech, Wuhan, China). Ultimately, an improved chemiluminescence detection technique (Peiqing Technology, Shanghai, China) was used to visualize protein bands. ImageJ ((x64)1.8.0, National Institutes of Health, Bethesda, MD, USA) was used to quantify the protein band density.

### 2.10. Statistical Analysis

GraphPad Prism software, version 8 (GraphPad Software, San Diego, CA, USA), was used to analyze all the data, which were represented as means ± SEM. Unpaired Student’s *t*-test, one-way analysis of variance, and Dunnett’s multiple comparisons test were used to accomplish the statistical evaluation of the data. Statistical significance was defined as a two-sided value of *p* < 0.05.

## 3. Results

### 3.1. Clinical and Laboratory Characteristics of the Study Population

Table 1 shows a comparison of the demographic and laboratory characteristics of the women evaluated with and without EOPE. Table 1 indicates that there was no statistically significant difference between the two groups’ age, gravidity, or gestational week at the time of blood collection. The SBP, DBP, BMI, and PSG1 levels were statistically different between the two groups. Compared to the EOPE group, the control group’s SBP, DBP, and BMI were all lower. However, as illustrated in Figure 1, the PSG1 level of the EOPE women was considerably lower than that of the control group. 

### 3.2. Effect of PSG1 on Migration and Proliferation of HTR-8/SVneo Cells

In the wound healing experiment, compared with the migration distance in the control group, the migration distance of cells was increased after the PSG1 treatment (Figure 2A). The cell migration rate data showed that the 0.1 and 0.5 μg/mL PSG1 treatments both significantly promoted cell migration ability, especially in the 0.1 μg/mL PSG1 group (Figure 2B). Using the CCK-8 assay, the proliferation of the HTR-8/SVneo cells was determined. Both the 0.1 and 0.5 μg/mL PSG1 treatments markedly increased HTR-8/SVneo proliferation in comparison to the control group (Figure 2C). Similarly, the proliferation in the treatment group with 0.1 μg/mL of PSG1 was stronger. There was no statistical difference in cell migration and proliferation between the concentrations of 0.1 and 0.5 μg/mL of PSG1. Meanwhile, neither the 0.1 nor 0.5 μg/mL PSG1 treatments in a serum-free basic culture medium had an effect on the HTR-8/SVneo cell proliferation in comparison to the control group (Figure 2D). These data indicate that PSG1 may promote the migration and proliferation of the HTR-8/SVneo cells and that cell proliferation did not affect the result of cell migration in the serum-free basic culture medium.

### 3.3. PSG1-Induced Transcriptome Profile Change in HTR-8/SVneo Cells

To determine the potential molecular mechanisms related to PSG1’s effect on HTR-8/SVneo cells, RNA sequencing (RNA-seq) technology was used to examine the changes in gene expression. Based on the above results, it was found that 0.1 μg/mL of PSG1 was enough to induce changes in cell viability and migration. Therefore, we chose a 0.1 μg/mL concentration to treat the HTR-8/SVneo cells and then performed an RNA-seq experiment. As demonstrated in the volcanic map and statistical chart (Figure 3A), we detected 122 genes, including 63 upregulated genes and 59 downregulated genes. When comparing the PSG1 treatment group to the control group, there was a multiple change of ≥2. Among the DEGs, 40 protein-encoded RNAs, four long non-coding RNAs (lncRNA), and 26 transcripts of nonsense-mediated decay were identified (Figure 3B). In addition, significant alterations in the expression profiles of genes in the HTR-8/SVneo cells are displayed on the heat map (Figure 3C). These findings indicate that the PSG1 treatment affected the transcripts of the HTR-8/SVneo cells.

### 3.4. Effect of PSG1 on Expression of Orai1 in HTR-8/SVneo Cells

According to the sequencing data, the PSG1 protein treatment considerably increased the Orai1 transcriptome expression in the HTR-8/SVneo cells. We used Western blotting and RT-qPCR to confirm this result. Orai1’s mRNA and protein expression levels were considerably higher after a 24 h PSG1 protein treatment compared to the control group (Figure 4A–C). The experimental results and sequencing data were consistent, indicating a significant increase in the expression level of Orai1 in the HTR-8/SVneo cells after incubation with PSG1. Moreover, in the Ca^2+^ imaging experiments, we also confirmed that the 0.1 μg/mL PSG1 treatment for 24 h significantly enhanced SOCE, and MRS1845, as an Orai1 inhibitor, markedly inhibited SOCE in both groups with and without PSG1 (Figure 4D,E). These results indicate that PSG1 may promote Orai1-mediated SOCE in HTR-8/SVneo cells.

### 3.5. Inhibition of Orai1 Reduced PSG1-induced HTR-8/SVneo Cell Migration

Next, we used a wound healing experiment to investigate how Orai1 affects cell migration. Compared with the control group, the migration rate was significantly increased in the PSG1-treated HTR-8/SVneo cells but was significantly inhibited by MRS1845 both with and without PSG1 treatment (Figure 5A,B). These results indicate that PSG1 may promote cell migration by enhancing the expression of Orai1 in HTR-8/SVneo cells.

### 3.6. Role of Akt in PSG1-Increased HTR-8/SVneo Cell Migration

According to several reports, regulating the migration and metastasis of different cancer cells is significantly influenced by the activation of Akt [25,26]. We employed Western blotting to assess the expression levels of total Akt and p-Akt to clarify the mechanism by which PSG1 controls HTR-8/SVneo cell migration through the Orai1 channel. In the cell migration experiments, the treatment with MRS1845 or ZSTK474 (an inhibitor of the Akt pathway) significantly inhibited PSG1-increased migration in HTR-8/SVneo cells (Figure 6A,B). As shown in Figure 6C,D, compared with the control group, PSG1 significantly promoted Akt phosphorylation, while MRS1845 significantly inhibited Akt phosphorylation with or without PSG1 treatment, and there was no significant difference in the MRS1845-caused inhibition effect between groups with or without PSG1. In addition, PSG1-increased p-Akt expression was also inhibited by ZSTK474. Therefore, these results indicate that PSG1 enhanced HTR-8/SVneo cell migration, potentially through the Orai1/Akt signaling pathway (Figure 7).

## 4. Discussion

PE affects 2% to 8% of pregnancies globally, with severe cases resulting in maternal and fetal mortality. The International Society for the Study of Hypertension in Pregnancy (ISSHP) defines PE as gestational hypertension accompanied by ≥1 of the following new-onset conditions at or after 20 weeks of gestation: proteinuria, other maternal organ dysfunction, and uteroplacental dysfunction [27,28,29]. Pregnant women are particularly affected by EOPE. Therefore, understanding the pathogenesis of EOPE is crucial. Previous studies have shown that impaired maternal spiral artery remodeling is a pathological feature of preeclampsia [30,31]. PSG1 serum levels were observed to be lower in individuals with EOPE in a previous study [32]. Consistent with this conclusion, we found that the serum PSG1 levels were lower in the pregnant women with EOPE than in the control group. The serum PSG1 concentration measured in the present study is similar to reports from other groups, ranging from 120 to 420 μg/mL [33,34], while some other studies also reported that the serum concentration of PSG1 is much lower than our data, ranging from tens of ng/mL to tens of μg/mL [32,35,36,37]. We speculate that this difference may be due to the differences in experimental procedures and reagent kits. PSG1 is a significant pregnancy-associated protein produced by the placenta and is intimately linked to the preservation of a healthy pregnancy. Thus, we hypothesize that preeclampsia may be associated with this decline in PSG1 levels.

PSG1 has been demonstrated in numerous studies to stimulate cell invasion, migration, and proliferation and inhibit inflammatory factors [38,39]. The mechanism by which PSG1 impacts the function of human trophoblast cells remains unknown. For this reason, we used HTR-8/SVneo cells to explore the effect of PSG1 on trophoblast cell migration and proliferation, as well as its potential mechanisms. Our main findings are as follows: (1) PSG1 stimulated HTR-8/SVneo cell growth and migration. (2) The RNA seq analysis showed that, compared with the control, the HTR-8/SVneo cells treated with PSG1 had 122 DEGs, of which 59 were downregulated and 63 were upregulated. The KEGG pathway enrichment analysis revealed that the PSG1-treated group also had increased Orai1 gene expression. (3) The expression levels of the Orai1 protein and SOCE activity in the HTR-8/SVneo cells treated with PSG1 significantly increased, while the inhibition of the Orai1 channel in the cells weakened SOCE activity and cell migration. (4) In the HTR-8/SVneo cells, PSG1 increased the expression of the p-Akt protein, and wound healing was significantly inhibited by applying the Orai1 inhibitor MRS1845 and/or an inhibitor of the Akt signal pathway. In summary, these data indicate that PSG1 increased the expression of Orai1 and SOCE activity in the HTR-8/SVneo cells, leading to an increase in p-Akt expression, thereby regulating the migration of human trophoblasts (Figure 7).

In many biological events, such as those that control cell migration, proliferation, and other processes, Ca^2+^ is a crucial regulatory component. An essential mechanism for maintaining intracellular Ca^2+^ homeostasis and Ca^2+^ signaling regulation is SOCE. Numerous investigations have demonstrated the involvement of SOCE-related proteins, such as Orais and STIM1, in signaling pathways associated with both normal and pathological processes. For example, Orai1-mediated SOCE is involved in the formation and development of pulmonary hypertension [40,41,42]. In the present study, we found that PSG1 at a concentration of 0.1 μg/mL had the ability to increase the Orai1 protein expression in the HTR-8/SVneo cells, ultimately promoting the migration of the HTR-8/SVneo cells. Meanwhile, SOCE in the HTR-8/SVneo cells also increased with an increment in Orai1 protein expression. On the contrary, after inhibiting the function of the Orai1 channel, the promoting effect of PSG1 on SOCE and migration in the cells was weakened. These findings suggest that Orai1 may be one of the potential mediators in the HTR-8/SVneo cell migration induced by PSG1. 

The Akt pathway’s involvement in preeclampsia pathogenesis has been demonstrated in many studies [43,44] and is also associated with the abnormal development of placental blood vessels [45]. Orai1 can regulate the Akt signaling pathway. Consequently, PSG1 may be involved in the Orai1/Akt pathway-mediated pathophysiology of EOPE. Our findings support this hypothesis by showing that the addition of the PSG1 recombinant protein greatly boosted the levels of Orai1 and p-Akt in the HTR-8/SVneo cells, as well as the cells’ capacity for migration. MRS1845 is a selective Orai1 channel blocker [46]. The treatment with MRS1845 inhibited the Orai1 channel and reduced the expression of p-Akt and the cell migration rate in the HTR-8/SVneo cells. Radoslavova et al. also reported that Orai1 can mediate Akt activation and promote pancreatic stellate cell migration in pancreatic ductal adenocarcinoma [47]. Therefore, our study suggests that PSG1 may regulate trophoblast migration through the Orai1/Akt pathway.

## 5. Conclusions

In summary, the PSG1 protein is found in reduced concentrations in the serum of pregnant women with EOPE. Additionally, PSG1 stimulates the migration of HTR-8/SVneo cells via the Orai1/Akt pathway; therefore, reduced PSG1 secretion may cause insufficient trophoblast migration and remodeling of spiral arteries and induce the occurrence and development of EOPE. This work shows the potential of PSG1 as a prospective target for the treatment of EOPE.

## Figures and Tables

**Figure 1 biomolecules-14-00293-f001:**
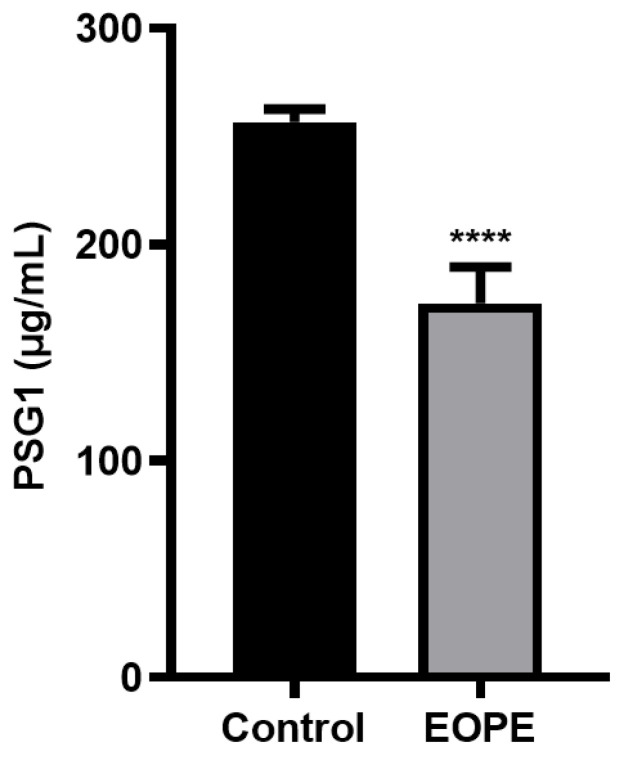
Protein level of pregnancy-specific beta-1-glycoprotein 1 (PSG1) in serum. PSG1 concentration in the serum of patients with early-onset preeclampsia (EOPE) and control groups. The data are displayed as mean ± SEM; **** *p* < 0.0001 for EOPE (*n* = 7) vs. control (*n* = 10).

**Figure 2 biomolecules-14-00293-f002:**
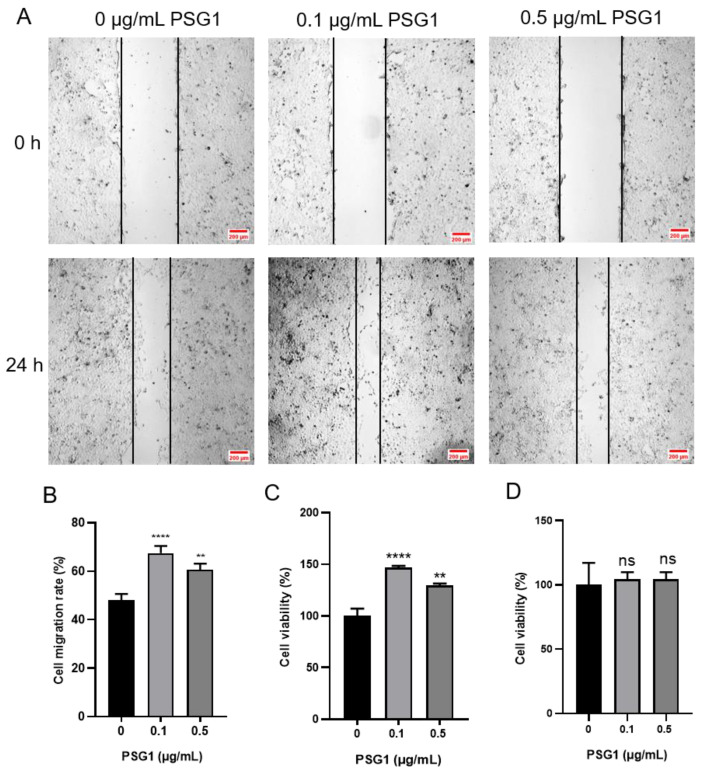
Effect of pregnancy-specific beta-1-glycoprotein 1 (PSG1) on migration and viability of HTR-8/SVneo cells. (**A**) Illustrative pictures displaying HTR-8/SVneo cell migration after being exposed to 0, 0.1, and 0.5 μg/mL of PSG1 for 24 h. (**B**) Summary data indicating the percentage of cells that migrated throughout a 24 h period. (**C**) Summary data showing the survival of HTR-8/SVneo cells after being exposed to 0, 0.1, and 0.5 μg/mL of PSG1 for 24 h. (**D**) Summary data showing the proliferation of HTR-8/SVneo cells after being exposed to 0, 0.1, and 0.5 μg/mL of PSG1 for 24 h in a serum-free basic culture medium. The scale bar in (**A**) is 200 μm. The data are displayed as mean ± SEM for *n* = 4–5. Dunnett’s multiple comparisons test was used to examine the data via a one-way analysis of variance (** *p* < 0.01, **** *p* < 0.0001 vs. 0 μg/mL of PSG1, ns, not significant).

**Figure 3 biomolecules-14-00293-f003:**
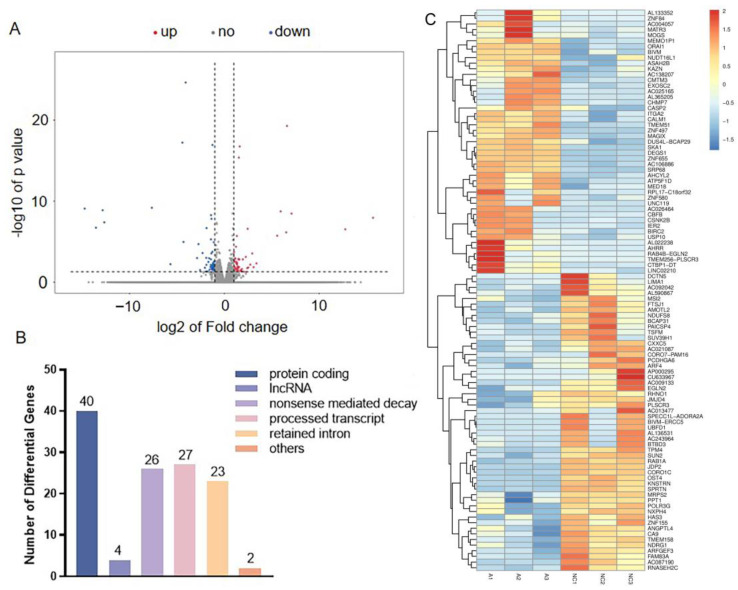
Effect of pregnancy-specific beta-1-glycoprotein 1 (PSG1) on transcriptome of HTR-8/SVneo cells. (**A**) Volcano map of differentially expressed genes (DEGs). (**B**) The number of transcripts with differential expression. (**C**) Heat map of genes with differential expression. Genes indicated by blue color are those with a low expression, and genes indicated by red color are those with a high expression. DEGs were defined as an absolute fold change ≥ 2 and an adjusted *p* < 0.05. The PSG1-treated group (0.1 μg/mL for 24 h) is indicated by A1-3, while NC1-3 indicates the control group.

**Figure 4 biomolecules-14-00293-f004:**
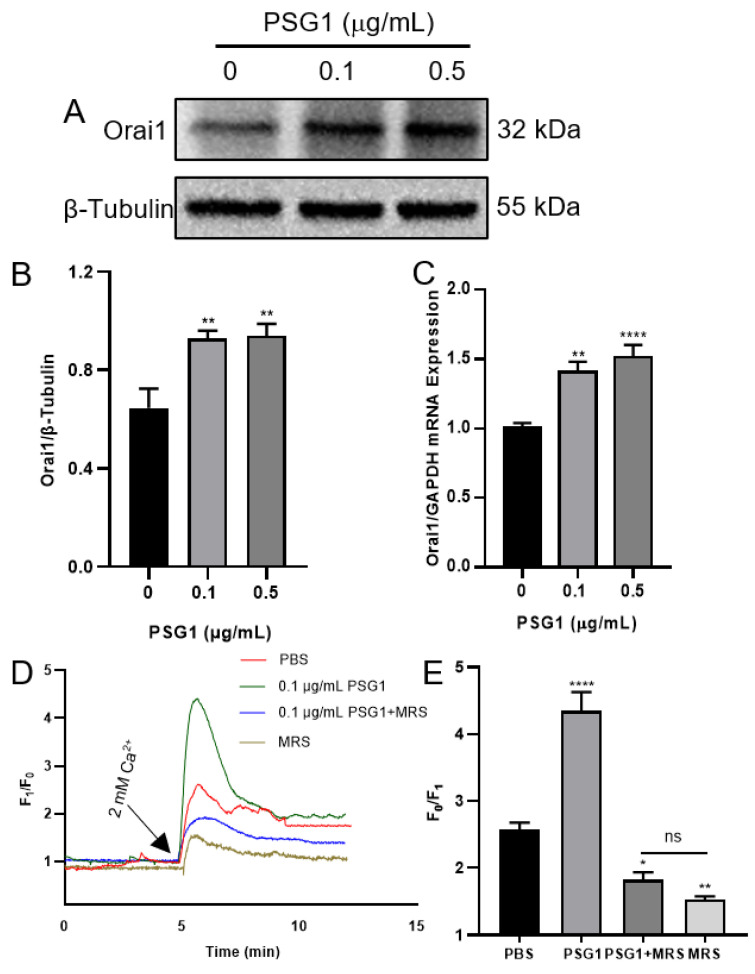
Effect of pregnancy-specific beta-1-glycoprotein 1 (PSG1) on Orai1 expression and store-operated Ca^2+^ entry (SOCE) in HTR-8/SVneo cells. After being treated with 0, 0.1, and 0.5 μg/mL of PSG1 for 24 h, the Orai1 protein expression in HTR-8/SVneo cells is demonstrated by representative images (**A**) and summary data (**B**). The loading control was the β-Tubulin protein. Following a 24 h treatment with 0, 0.1, and 0.5 μg/mL of PSG1, qPCR assays (**C**) were used to detect the mRNA levels in HTR-8/SVneo cells. The expression of messenger RNA was compared to that of GAPDH. The SOCE in HTR-8/SVneo cells was triggered by thapsigargin (TG), as demonstrated by representative traces (**D**) and summary data (**E**) following treatment with phosphate-buffered saline (PBS), PSG1 (0.1 μg/mL), PSG1 + 10 μM of MRS1845, or MRS1845 only for 24 h. Once the intracellular Ca^2+^ stores were depleted through treatment with 2 μM of TG for 10 min in a Ca^2+^-free medium, the addition of 2 mM of Ca^2+^ to the medium induced SOCE. Data are shown as the mean ± SEM; *n* = 5–7. * *p* < 0.05, ** *p* < 0.01, **** *p* < 0.0001 vs. PBS analyzed via one-way analysis of variance followed by Dunnett’s multiple comparisons test; ns, not significant. Original figures can be found in Appendix A.

**Figure 5 biomolecules-14-00293-f005:**
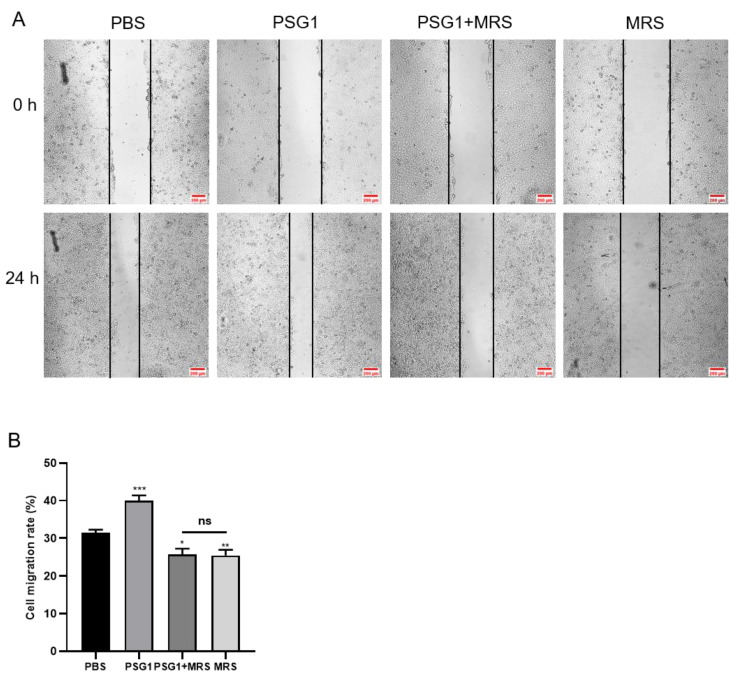
Orai1 blocker inhibited promotional effect of pregnancy-specific beta-1-glycoprotein 1 (PSG1) on the migration of HTR-8/SVneo cells. (**A**) Representative images of HTR-8/SVneo cell migration after treatment of cells with phosphate-buffered saline (PBS), 10 μM of MRS1845, and/or 0.1 μg/mL of PSG1. (**B**) Quantitative data showing the cell migration rate of HTR-8/SVneo cells treated with PBS, 10 μM of MRS1845, and/or 0.1 μg/mL of PSG1. The scale bar in (**A**) is 200 μm. Data are shown as the mean ± SEM; *n* = 5. * *p* < 0.05, ** *p* < 0.01, *** *p* < 0.001 vs. PBS; ns, not significant.

**Figure 6 biomolecules-14-00293-f006:**
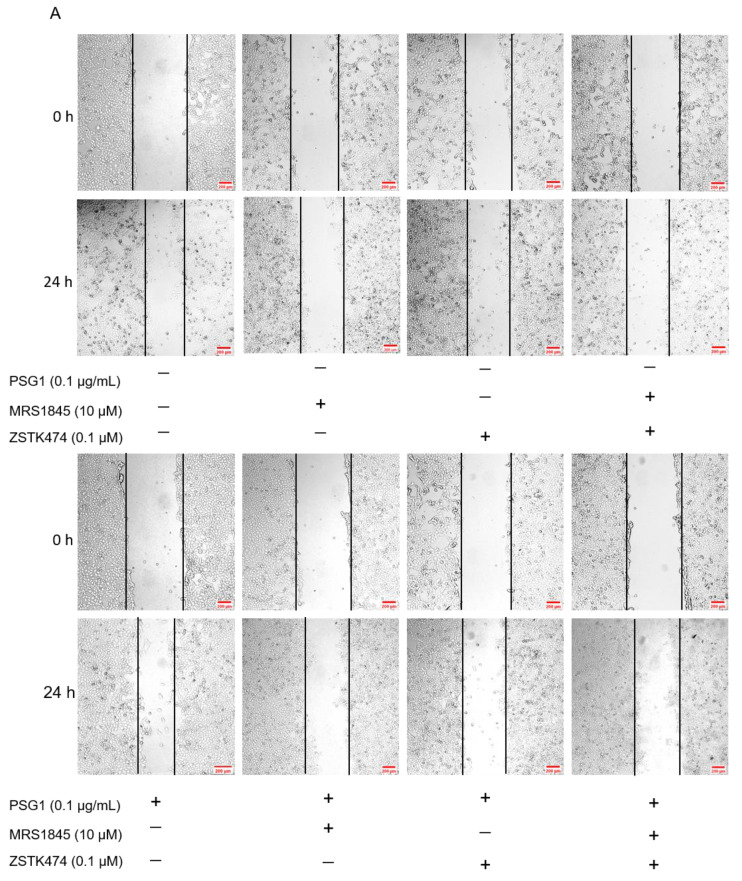
Effect of pregnancy-specific glycoprotein 1 (PSG1) on migration of HTR-8/SVneo cells through the Orai1/Akt signaling pathway. (**A**) Representative images of HTR-8/SVneo cell migration after treatment of cells with 10 μM of MRS1845, 0.1 μM of ZSTK474, and/or 0.1 μg/mL of PSG1. (**B**) Summary data showing the percentage of cells that migrated during 24 h. (**C**,**D**) Representative images and summary data showing the effect of 10 μM of MRS1845, 0.1 μM of ZSTK474, and/or 0.1 μg/mL of PSG1 on Akt/p-Akt expression in HTR-8/SVneo cells. GAPDH was used as a loading control. The scale bar in (**A**) is 200 μm. Data are shown as the mean ± SEM; *n* = 3–4. * *p* < 0.05, ** *p* < 0.01, **** *p* < 0.0001 vs. negative control (−), ^#^
*p* < 0.05, ^###^
*p* < 0.001, ^####^
*p* < 0.0001 vs. PSG1 analyzed via one-way analysis of variance followed by Dunnett’s multiple comparisons test. Original figures can be found in Appendix A.

**Figure 7 biomolecules-14-00293-f007:**
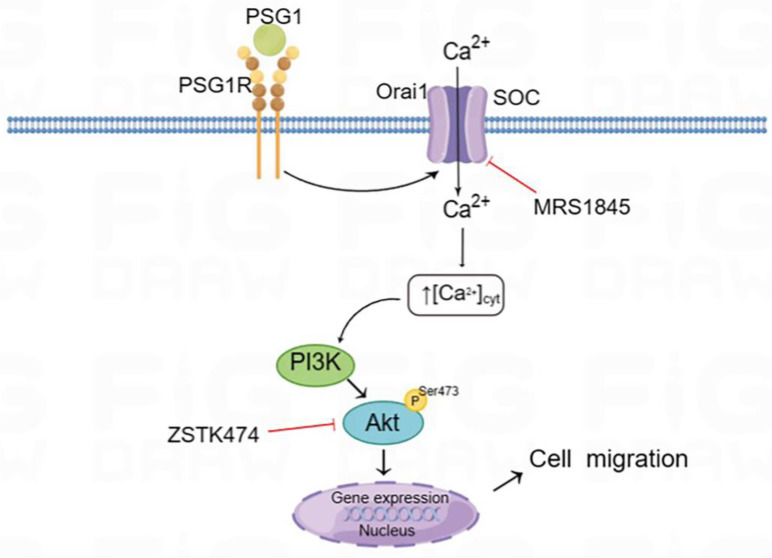
Schematic diagram of the Orai1-mediated store-operated Ca^2+^ entry (SOCE) signal transduction pathway in HTR-8/SVneo cells after PSG1 treatment. Binding of PSG1 to its receptor (PSG1R) activated the expression of Orai1 and increased SOCE, leading to the activation of Akt signaling and increased cell migration.

**Table 1 biomolecules-14-00293-t001:** Comparison of obstetric characteristics and PSG1 levels.

Variables	Control (*n* = 10)	Early-Onset Preeclampsia (*n* = 7)	*p*
Age, year	30.60 ± 4.17	31.43 ± 4.61	0.705
Gravidity, times	2.90 ± 1.60	3.43 ± 0.79	0.433
Gestational age, weeks	33.01 ± 1.50	32.55 ± 1.23	0.519
BMI, kg/m^2^	25.21 ± 3.96	29.51 ± 2.02	**0.019**
SBP, mmHg	103.60 ± 4.27	145.57 ± 9.00	**<** **0.0001**
DBP, mmHg	69.70 ± 6.77	95.00 ± 9.02	**<** **0.0001**
PSG1, μg/mL	256.53 ± 18.95	172.86 ± 44.53	**<** **0.0001**

Bold data showing *p*-value < 0.05 and significant changes.

## Data Availability

Data will be made available upon request to the corresponding author.

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
