# Peer review of "Pregnancy-Specific Beta-1-Glycoprotein 1 Increases HTR-8/SVneo Cell Migration through the Orai1/Akt Signaling Pathway"

_biomolecules, 2024, doi:10.3390/biom14030293_

Round 1
Reviewer 1 Report
Comments and Suggestions for Authors
The manuscript named “Pregnancy-specific beta-1-glycoprotein 1 increases HTR-8/SVneo cell migration through Orai1/Akt signaling pathway” presents interesting results. It shows how PSG1 is involved in regulation of trophoblast migration and proposes a molecular mechanism through which PSG1 acts in trophoblast cells as well as possible association of this molecule to the preeclampsia pathophysiology. Although research is relevant, English in this manuscript is not good enough. I would recommend authors to carefully rewrite the manuscript to improve English so good results that they are presenting could be better appreciated. Like this, manuscript is not acceptable for publishing.
Also, there are other aspects that could be improved.
- How you choose PSG1 concentrations for experiments? What was the source of PSG1 protein - recombinant protein or isolated from serum? It should be written in Material and Methods section.
- How come the viability of untreated control cells is higher than 100 % (Figure 2C)? Please check the results and the graph.
- How are you sure that PSG1 specifically stimulated HTR-8/SVneo migration and that your results are not a consequence of the stimulated cell proliferation by PSG1 (Figure 2C)? Did you check how MRS1845 and ZSTK474 affect HTR-8/SVneo cells?
- Figure 4A: PSG1 treatment concentrations are not designated to the bands on the representative Western blot image. Please add treatment concentration.
- What does the fourth band in the original WB 1 images represent?
- Figure 4B and C: Please check graphs and y axes signs. It looks like graphs are the same while in the figure legend is written that one should represent PSG1 mRNA and other PSG1 protein expression.
- Figure 5A: Cells are not visible on the images for 0h. Please provide better images.
- How do you mean “partially inhibited promotional effect”? MRS1845 significantly inhibited HTR-8/SVneo migration both alone and in combination with PSG1 treatment.
- L324: “leading to the activation of Akt signaling” instead of “expression of Akt signaling”.
10. L34-36 and L364-366: The statement is too strong. Indeed, it was shown that PSG1 affects trophoblast migration acting through Orai1/Akt signaling pathways, but it is one of the mechanisms possibly involved in EOPE pathophysiology, not the key mechanism.
- L31-32: Results confirmed ...not indicated...since you and other authors have already published that PSG1 serum levels were decreased in EOPE women.
- L45-46: You’ve written that “PSG1 plays an important role...”. Could you be more specific about that? Important role in which processes?
- L46-47: The sentence is not clearly written. “The primary cause of maternal and perinatal incidence rate...”. Incidence of which disease? Did you maybe mean that EOPE is a major cause of maternal and perinatal morbidity and mortality? Please rephrase this sentence.
- L51-55: Please rephrase this sentence. I would recommend dividing it into 2 sentences to be more clear and easier to follow. The first one should be about how insufficient trophoblast invasion causes impaired remodeling of spiral arteries, pointing out the fact that spiral arteries in the myometrium are not transformed in preeclampsia as much as ones in normal pregnancies if that was what you wanted to point out. The second sentence should be about the consequences of impaired remodeling of spiral arteries.
- L57: Did you mean PSG1 or PSG protein family? Please be more specific.
- L59-60: This sentence is repeating the information from the previous one (L57-59) and it is not precise. What did you mean by “one of the most abundant fetal proteins”? In maternal serum? It is not clear what you meant.
- L63: Please check reference 12. I couldn’t find any information about the association of PSG1 with recurrent miscarriage in the cited article. Also, there is no reference about the association of PSG1 with fetal growth restriction.
- L69-70: SOCE is the primary pathway for Ca2+ entry into the cell
- L73-74: What did you mean by “different cell migration”?
- L177: “chorionic” is unnecessary in this sentence.
- L135: 10 ml or 10 μl?
- L140-141: What 50-60% affinity means in this context? Did you mean: HTR-8/SVneo cells were grown on glass coverslips until they reached 50-60% confluence? Please rephrase this sentence.
- L144: What did you mean by “final concentration of 60%”? Final concentration of what?
- L229-230: How did you measure “proportion of cells that moved…”? What are units on y axis in Figure 2B?
Comments on the Quality of English Language
Many sentences are not clear, complete or precise. They are not written in the way characteristic for the English language (i.e.L46-47, L51-55, L57-58, L61-62, L67-68, L70-71, L76-77…L332-334…). Some nouns are written in singular although they should be written in plural (i.e. L52: uterine spiral artery; L53: transformation of spiral artery). Sentences following each other are repeating information (L57-59, L370-375). Incorrect terms are used throughout this manuscript and should be corrected:
L43: “bodily systems” should be replaced with “organ systems”
L44: Syncytiotrohpoblast is syncytium therefore there are no separated cells. Please replace “syncytiotrohpoblast cells” with “syncytiotrohpoblast”.
L51: Correct term is “trophoblast cells” not “trophoblastic cells”. Please replace “trophoblastic cells” with “trophoblast cells”.
L121: Instead “...cell aggregation rate reached…” more suitable is “...when cells reached 80-90%…”
L124: “Plate” instead of “board”
L186: “after being sealed in 5% skim milk”?
L334-335: “protein particular to the pregnancy”; “prenatal trophoblast”
L23: Infiltration ability – I would recommend to use more suitable term invasion ability;
L30: Add ,respectively at the end of the sentence;
And so on.
Author Response
Point-by-point Response
General response: Thank you very much for your comments and professional advice. We have carefully evaluated the reviewers’ critical comments and thoughtful suggestions. These opinions help us to improve the manuscript. All changes in the manuscript are marked in red. In addition, the manuscript had be reviewed and edited by language services of MDPI. We hope that revised manuscript has addressed all criticisms raised by the editor and the reviewers and that the manuscript is now suitable for publication.
Reviewers' Comments to Author:
Reviewer #1:
- The manuscript named “Pregnancy-specific beta-1-glycoprotein 1 increases HTR-8/SVneo cell migration through Orai1/Akt signaling pathway” presents interesting results. It shows how PSG1 is involved in regulation of trophoblast migration and proposes a molecular mechanism through which PSG1 acts in trophoblast cells as well as possible association of this molecule to the preeclampsia pathophysiology. English in this manuscript is not good enough. I would recommend authors to carefully rewrite the manuscript to improve English so good results that they are presenting could be better appreciated. Like this, manuscript is not acceptable for publishing.
Answer: Thanks for your suggestion. The manuscript had be reviewed and edited by language services of MDPI. We hope that the manuscript is now suitable for publication.
- How you choose PSG1 concentrations for experiments?
Answer: Thank you very much for your question. Cell viability assays were carried out in the pre-experiment at three different PSG1 concentrations: 0, 0.1 and 0.5 μg/mL. It was found that 0.5 μg/mL PSG1 was enough to induce cell viability change, and 0.5 μg/mL concentration was chosen for our experiment. The pre-experiment results are displayed in the following figure:
- What was the source of PSG1 protein - recombinant protein or isolated from serum? It should be written in Material and Methods section.
Answer: Thank you very much for your careful check. The Recombinant human PSG1 product was purchased from R&D Systems under item number 9334-P1-050. The source of recombinant human PSG1 protein has been added to the Cell Culture section of the Material and Methods.
- How come the viability of untreated control cells is higher than 100 % (Figure 2C)? Please check the results and the graph.
Answer: Thank you very much for your criticism. We are sorry for the unreasonable results caused by the incorrect calculation method. We check our data again and corrected the analysis method to adjust the initial viability rate into 100%. Please check Figure 2C again.
- How are you sure that PSG1 specifically stimulated HTR-8/SVneo migration and that your results are not a consequence of the stimulated cell proliferation by PSG1 (Figure 2C)?
Answer: Thank you very much for your question. In order to minimize the impact of cell proliferation on the migration experiments, the scratched cells were cultured in a serum-free basic culture medium. Therefore, the cell proliferation was largely reduced in this medium. We added this detailed information into the Method section. Please check it again.
- Did you check how MRS1845 and ZSTK474 affect HTR-8/SVneo cells?
Answer: Thank you very much for your question. In Figures 6A-B, the results showed that both MRS1845 alone and ZSTK474 alone inhibited the migration ability of HTR-8/SVneo. Please check them again.
- Figure 4A: PSG1 treatment concentrations are not designated to the bands on the representative Western blot image. Please add treatment concentration.
Answer: Thank you very much for your valuable suggestion. The PSG1 treatment concentrations was added to the bands on representative protein blot images in this revised manuscript (please see Figure 4A).
- What does the fourth band in the original WB 1 images represent?
Answer: Thank you very much for your question. The fourth band was to observe the effect of PSG9 on cells, as it was another study and the results were temporarily not suitable for adding into this study.
- Figure 4B and C: Please check graphs and y axes signs. It looks like graphs are the same while in the figure legend is written that one should represent PSG1 mRNA and other PSG1 protein expression.
Answer: Thank you very much for your criticism. After careful check, due to our mistake, the protein statistical figure was wrongly displayed as the mRNA statistical figure. We revised it in this version of manuscript. Please check Figure 4C again.
- Figure 5A: Cells are not visible on the images for 0h. Please provide better images.
Answer: Thank you very much for your constructive suggestion. We made the corresponding modifications to make the images clearer in this revised version. Please check Figure 5A again.
- How do you mean “partially inhibited promotional effect”? MRS1845 significantly inhibited HTR-8/SVneo migration both alone and in combination with PSG1 treatment.
Answer: Thank you very much for your criticism. After careful verification, it was found that the word "partially" was used improperly. In this revised manuscript, we deleted this word.
- L324: “leading to the activation of Akt signaling” instead of “expression of Akt signaling”.
Answer: Thank you very much for your valuable suggestion. We have revised this sentence based on your suggestion, changing "expression of Akt signing" to "leading to the activation of Akt signing".
- L34-36 and L364-366: The statement is too strong. Indeed, it was shown that PSG1 affects trophoblast migration acting through Orai1/Akt signaling pathways, but it is one of the mechanisms possibly involved in EOPE pathophysiology, not the key mechanism.
Answer: Thank you very much for your kindly suggestions. Accordingly, we reduced our tune and revised two statements into “Our findings revealed one of the mechanisms possibly involved in EOPE pathophysiology, which was that downregulated PSG1 may reduce the Orai1/Akt signaling pathway, thereby inhibiting trophoblast migration” “These findings suggested that Orai1 may be one of potential mediators in the HTR-8/SVneo cell migration induced by PSG1” Please check both them again.
- L31-32: Results confirmed ...not indicated...since you and other authors have already published that PSG1 serum levels were decreased in EOPE women.
Answer: Thank you very much for your valuable suggestion. We have revised "indicated" to "confirmed".
- L45-46: You’ve written that “PSG1 plays an important role...”. Could you be more specific about that? Important role in which processes?
Answer: Thank you very much for your valuable suggestion. We revised the sentence as follows: "PSG1 plays a potential role in placental vascular morphology and is closely related to abnormal gestations and intrauterine fetal growth". Please check it again.
- L46-47: The sentence is not clearly written. “The primary cause of maternal and perinatal incidence rate...”. Incidence of which disease? Did you maybe mean that EOPE is a major cause of maternal and perinatal morbidity and mortality? Please rephrase this sentence.
Answer: Thank you very much for your careful check. We have revised this sentence to more specific content as follows:“Preeclampsia affects 2% to 8% of pregnancies worldwide and causes significant maternal and perinatal morbidity and mortality. Early-onset preeclampsia (EOPE), in particular, confers a high risk of life-threatening maternal complications and fetal demise”. Please check it again.
- L51-55: Please rephrase this sentence. I would recommend dividing it into 2 sentences to be more clear and easier to follow. The first one should be about how insufficient trophoblast invasion causes impaired remodeling of spiral arteries, pointing out the fact that spiral arteries in the myometrium are not transformed in preeclampsia as much as ones in normal pregnancies if that was what you wanted to point out. The second sentence should be about the consequences of impaired remodeling of spiral arteries.
Answer: Thank you very much for your constructive suggestions. Following your suggestion, we have revised this sentence as follows: “Although the exact cause is not yet clear, it is widely believed that insufficient invasion of the trophoblast is one of the important pathological factors in preeclampsia. During the development of the placenta, the migration and invasion abilities of trophoblast cells are reduced, and the recast of trophoblast cells in the uterine spiral artery is impaired, leading to placental ischemia and hypoxia, causing a series of oxidative stress reactions, thereby causing several symptoms of preeclampsia”. Please check it again.
- Did you mean PSG1 or PSG protein family? Please be more specific.
Answer: Thank you very much for your question. After careful check, we have revised the sentence into "Pregnancy specific glycoproteins (PSGs) share strong sequence similarity with cancer associated antigen (CEA)". Please check it again.
- L59-60: This sentence is repeating the information from the previous one (L57-59) and it is not precise. What did you mean by “one of the most abundant fetal proteins”? In maternal serum? It is not clear what you meant.
Answer: Thank you very much for your questions. We have removed duplicate information and revised the sentence as follows: “Pregnancy-specific glycoproteins (PSGs) share strong sequence similarity with carcinoembryonic antigen (CEA) family dedicated to pregnancy and are mainly secreted by trophoblast cells. PSGs are the most abundant proteins in the maternal blood-stream during late pregnancy”. Please check it again.
- L63: Please check reference 12. I couldn’t find any information about the association of PSG1 with recurrent miscarriage in the cited article. Also, there is no reference about the association of PSG1 with fetal growth restriction.
Answer: We sincerely appreciate the valuable suggestion. We carefully reviewed the literature and removed inappropriate citations. According to the new reference 13, PSG1 is associated with recurrent miscarriage. Reference 2 is about the relationship between PSG1 and fetal growth restriction. Please check them again.
- L69-70: SOCE is the primary pathway for Ca2+ entry into the cell
Answer: Thank you very much for your careful check. We have revised this sentence according to your suggestion.
- L73-74: What did you mean by “different cell migration”?
Answer: Sorry for the writing problems. We have corrected it into "migration in different cells".
- L177: “chorionic” is unnecessary in this sentence.
Answer: Thank you very much for your careful check. We have removed this word in this revised manuscript.
- L135: 10 ml or 10 μl?
Answer: We were really sorry for our mistakes. We have corrected it into 10 μl.
- L140-141: What 50-60% affinity means in this context? Did you mean: HTR-8/SVneo cells were grown on glass coverslips until they reached 50-60% confluence? Please rephrase this sentence.L144: What did you mean by “final concentration of 60%”? Final concentration of what?
Answer: Thank you very much for your valuable suggestion. We revised the sentence into “HTR-8/SVneo cells were grown on glass coverslips until they reached 50-60% confluence. To load the Ca2+ selective fluorescent indicator, the cells were incubated with 6 μmol/L Fluo-8 AM for 30 min at 37°C prior to the experiment” to make it readable.
- L229-230: How did you measure “proportion of cells that moved…”? What are units on y axis in Figure 2B?
Answer: We were really sorry for our mistakes. In this revised manuscript, we have corrected it into “Summary data indicating the percentage of cells that migrated throughout a 24 h period” and the units on y axis in Figure 2B have been added (Please see Figure 2B). Please check it again.
- Comments on the Quality of English Language.
Many sentences are not clear, complete or precise. They are not written in the way characteristic for the English language (i.e.L46-47, L51-55, L57-58, L61-62, L67-68, L70-71, L76-77…L332-334…). Some nouns are written in singular although they should be written in plural (i.e. L52: uterine spiral artery; L53: transformation of spiral artery). Sentences following each other are repeating information (L57-59, L370-375).
Answer: Thanks for your careful check. The manuscript had be reviewed and edited by language services of MDPI. We have carefully checked and corrected the issue that you pointed out, deleted and rewritten sentences accordingly.
- L43: “bodily systems” should be replaced with “organ systems”
Answer: Thank you very much for your careful check. We have replaced "boldly systems" with "organized systems" in this revised manuscript.
- L44: Syncytiotrohpoblast is syncytium therefore there are no separated cells. Please replace “syncytiotrohpoblast cells” with “syncytiotrohpoblast”.
Answer: Thank you very much for your careful check. We have replaced all the "syncytiotrophoblast cells" with "syncytiotrophoblast" in this revised manuscript.
30 .L51: Correct term is “trophoblast cells” not “trophoblastic cells”. Please replace “trophoblastic cells” with “trophoblast cells”.
Answer: Thank you very much for your careful check. We have replaced all the "trophoblastic cells" with "trophoblast cells" in this revised manuscript.
- L121: Instead “...cell aggregation rate reached…” more suitable is “...when cells reached 80-90%…”
Answer: Thanks for your careful check. We have rewritten the sentence according to your suggestion as follows: “The culture medium was changed every 2 days, and when the cells reached to 60%-80% ”.
- L124: “Plate” instead of “board”
Answer: Thanks for your careful check. We have replaced "board" with "plate".
- L186: “after being sealed in 5% skim milk”?
Answer: Thanks for your careful check. We have changed the sentence to "after being blocked in 5% skin milk for 2 h"
- L334-335: “protein particular to the pregnancy”; “prenatal trophoblast”
Answer: Thanks for your careful check. We have rewritten the sentence as follows: “PSG1 is a significant pregnancy-associated protein produced by the placenta”.
- L23: Infiltration ability – I would recommend to use more suitable term invasion ability;
Answer: Thank you very much for your careful check. We have changed "infiltration capability" to "invasion capability".
- L30: Add, respectively at the end of the sentence;
Answer: Thank you very much for your careful check. We have added “respectively” at the end of the sentence.
Reviewer 2 Report
Comments and Suggestions for Authors
The study is very interesting and generally well written. However, some points deserves to be improved. In particular:
Lines 48-55: It deserves to be mentioned that PE pregnancies are also charactherized by an increased oxidative stress (see PMID: 37296665). This is an important point to add since it can contribute to the systemic alteration found in the mother.
Line 156: the product code of the recombinant PSG1 must be reported
2.9. Western Blotting: did the authors block the membranes before the incubation with the primary antibodies?
3.1. Clinical and Laboratory Characteristics of Study Population: the sample size is too small to reach any conclusion. I understand that EOPE is a quire rare complication of pregnancy but authors could al least increase the number of the control samples.
Table 1: the statistically significant values should be written in bold.
Figure 2A, 5A and 6A: Images are too small and scale bars are missed
Figure 4A: Densitometric analysis is missed
Abbreviations must be written in full length when mentioned for the first time
Author Response
Point-by-point Response
General response: Thank you very much for your comments and professional advice. We have carefully evaluated the reviewers’ critical comments and thoughtful suggestions. These opinions help us to improve the manuscript. All changes in the manuscript are marked in red. In addition, the manuscript had be reviewed and edited by language services of MDPI. We hope that revised manuscript has addressed all criticisms raised by the editor and the reviewers and that the manuscript is now suitable for publication.
Reviewers' Comments to Author:
Reviewer #2:
The study is very interesting and generally well written. However, some points deserves to be improved. In particular:
- Lines 48-55: It deserves to be mentioned that PE pregnancies are also characterized by an increased oxidative stress (see PMID: 37296665). This is an important point to add since it can contribute to the systemic alteration found in the mother.
Answer: We sincerely appreciate the valuable suggestions. We carefully reviewed the literature and added a description of the involvement of oxidative stress response in the occurrence and development of PE in the revised manuscript, along with references. As follows:
Reference 5: Lisonkova S, Joseph KS. Incidence of preeclampsia: risk factors and outcomes associated with early-versus late-onset disease. Am J Obstet Gynecol. 2013; 209(6): 544.e1-544.e12. DOI:10.1016/j.ajog.2013.08.019
- Line 156: the product code of the recombinant PSG1 must be reported
Answer: Thank you very much for your suggestion. The Recombinant human PSG1 product was purchased from R&D Systems under item number 9334-P1-050. And the source of recombinant human PSG1 protein has been added to the Cell Culture section of the Material and Methods.
- Western Blotting: did the authors block the membranes before the incubation with the primary antibodies?
Answer: Thank you very much for your question. It is yes, the membranes were blocked in 5% skim milk for 2 h before primary antibodies were incubated. We added this information into the Method section. Please check it again.
- Clinical and Laboratory Characteristics of Study Population: the sample size is too small to reach any conclusion. I understand that EOPE is a quire rare complication of pregnancy but authors could at least increase the number of the control samples.
Answer: Thank you very much for your constructive suggestions. Indeed, due to the limited number of EOPE patients, it is very difficult to obtain suitable samples. In order to increase clinical data, we have made effort to increase the data in the control group according to your suggestion. In the revised manuscript, we increased the sample size of the control group in the Elisa experiment into 10 and made corresponding modifications in Table 1 and Figure 1 (Please see Table 1 and Figure 1).
- Table 1: the statistically significant values should be written in bold.
Answer: Thank you very much for your valuable suggestion. The statistically significant values in revised manuscript have been revised into bold.
- Figure 2A, 5A and 6A: Images are too small and scale bars are missed
Answer: Thank you very much for your criticism. We have adjusted the images make them more readable and added the scale bars in this revised manuscript.
- Figure 4A: Densitometric analysis is missed
Answer: We were really sorry for our mistakes. In this revised manuscript, the densitometric analysis is added. Please take a look at Figure 4B.
- Abbreviations must be written in full length when mentioned for the first time
Answer: We sincerely appreciate the valuable suggestions. All abbreviations in this revised manuscript have been provided in a full name when mentioned for the first time.

Reviewer 3 Report
Comments and Suggestions for Authors
Dear Authors,
my comments:
1. In my opinion extensive English editing is needed
2. All abbrevioations have to be desribed properly
3. Inreoduction line 47- I cannot agree that "primary" is a good word this sentence.
4. lines 328-329 - "terminal organ damage" not proper wrods to define PE, not full definition
Comments on the Quality of English LanguageDear Editor,
this article might be useful, but needs major revision... firstly English editing, secondly wrong definition of PE, that are serious mistakes.
Author Response
Point-by-point Response
General response: Thank you very much for your comments and professional advice. We have carefully evaluated the reviewers’ critical comments and thoughtful suggestions. These opinions help us to improve the manuscript. All changes in the manuscript are marked in red. In addition, the manuscript had be reviewed and edited by language services of MDPI. We hope that revised manuscript has addressed all criticisms raised by the editor and the reviewers and that the manuscript is now suitable for publication.
Reviewer #3:
- In my opinion extensive English editing is needed
Answer: Thank you very much for your valuable suggestion. The manuscript had be reviewed and edited by language services of MDPI. We hope that the manuscript is now suitable for publication.
- All abbreviations have to be described properly
Answer: We sincerely appreciate the valuable suggestions. All abbreviations in this revised manuscript have been provided in a full name when mentioned for the first time.
- Line 47- I cannot agree that "primary" is a good word this sentence.
Answer: Thank you very much for your criticism. We have revised the sentence as follows: “Preeclampsia affects 2% to 8% of pregnancies worldwide and causes significant maternal and perinatal morbidity and mortality”. Please check it again.
- Lines 328-329 - "terminal organ damage" not proper words to define PE, not full definition.
Answer: Thank you very much for your criticism. We have revised the definition of PE according to the guidelines as follows: “PE as gestational hypertension accompanied by ≥1 of the following new-onset conditions (such as proteinuria, other maternal organ dysfunction, and uteroplacental dysfunction) at or after 20 weeks of gestation”. Please check it again.

Round 2
Reviewer 1 Report
Comments and Suggestions for Authors
The authors accepted remarks about writing and quality of English and made improvement of the manuscript. Also, some technical details about graphs and images were improved. Nevertheless, writing still should be improved. Many sentences are not clear or precise, some words important for the statements are missing, etc. Furthermore, some of the methodological questions weren’t answered completely and some other questions appeared after repeated reviewing of the manuscript. These questions should be answered and listed language corrections together with others which I maybe missed should be incorporated it the manuscript before it will be suitable for the publishing.
1. How come gestational age at blood sampling for EOPE group was 33.21±1.97 (Table 1)? Does it mean that at least one of the pregnant women in this group was more than 34 weeks pregnant at the time of blood sampling? If that so, how come you enrolled that/those women into EOPE group since EOPE is defined by new-onset hypertension and proteinuria before 34th week of gestation? Please provide results for women diagnosed with EOPE and if possible, for more than four EOPE patients.
2. Serum PSG1 levels in healthy pregnant women was much higher in your measurements, (256.53±18.95 μg/mL) compared to other published data e.g. (10.1080/01443615.2019.1679734, 10.5935/1518-0557.20210068, 10.1081/prg-120016794, https://doi.org/10.3390/cells8111369) where PSG1 serum levels were at tens of μg/mL or even in ng/ml although healthy pregnant woman participants were in the similar gestational age at the time of blood sampling. Please comment on that.
3. You chose 0.1 and 0.5 μg/mL PSG1 for treatment of HTR-8/SVneo cells in your experiments. Can you explain on which bases you chose these concentrations? That should also be written in the manuscript. In your answer on the same question after first revision you said that you conducted cell viability test with those PSG1 concentrations and that you “found that 0.5 μg/mL PSG1 was enough to induce cell viability change and 0.5 μg/mL concentration was chosen for our experiment.” First, I couldn’t find the graph of pre-experiments you are referring to in your answer after first revision. Second, cell viability assay results from the manuscript (Figure 2C) show that already 0.1 μg/mL PSG1 increased HTR-8/SVneo cell proliferation significantly.
4. Next, why did you choose PSG1 concentrations that stimulate cell proliferation for wound healing assay instead of lower concentration which do not change cell proliferation? In this way, cell proliferation stimulation could interfere with the cell migration results. Looking at your results (Figure 2B and 2C), rates of PSG1 stimulation of HTR-8/SVneo cell migration are almost the same as cell proliferation stimulation rates for the same PSG1 concentrations. Also, in your answers after first revision you said: “In order to minimize the impact of cell proliferation on the migration experiments, the scratched cells were cultured in a serum-free basic culture medium.” Did you perform cell viability assays in the same conditions (0.1 and 0.5 μg/mL PSG1, serum-free basic medium, 24 h treatment)? If yes present results and comment on them. If not, I recommend checking if PSG1 affects cell proliferation in serum-free conditions.
5. Please discuss migration assay results in connection with cell proliferation results in manuscript also.
6. Please add why you chose 0.1 μg/ml PSG1 treatment for transcriptome analysis in connection with the other results in the manuscript.
7. Please write in more details how you treated cells for Ca2+ imaging experiments before adding Fluo-8 AM in Material and Methods section (2.6). How long was treatment with PSG1 and MRS1845?
8. Please add experimental procedures for treating cells with MRS1845 and ZSTK474 in Wound healing experiments and for Western blot experiments.
9. Please give a little bit more information in the Introduction section about EOPE and LOPE, the differences between these two PE subtypes regarding clinical manifestations and suggested pathophysiology. Discuss shortly why you focused on EOPE.
10. L42-72: I would recommend reorganizing this section in a way that in the first paragraph you present relevant data about preeclampsia as you started it and in the second paragraph to give relevant information about PSGs and PSG1. Regarding that, I would recommend moving sentence L46-47 to the second paragraph.
11. L30: Please add the word “trophoblast” before “proliferation and migration”.
12. L34-36: Please rephrase this sentence. The part: “MRS1845...also reducing the promoting effect of PSG1 via the Akt signaling pathway” is not clear.
13. L44-45: Please add the word “new-onset” before “hypertension and proteinuria” and add suitable reference for this statement.
14. L47: Please add the word “maternal” before “bloodstream.”
15. L54: Please use PE and other abbreviations throughout the manuscript after introducing them.
16. L54-58: Sentence is still not precise and clear enough. Please reorganize it this way: In PE, the migration and invasion abilities of trophoblast cells are reduced, and remodeling of the uterine spiral arteries is impaired, leading to placental ischemia and hypoxia, causing a series of oxidative stress reactions, thereby causing a series of several symptoms of PE.
17. L58: Please write “mechanisms” instead of “mechanism”.
18. L61: “Dedicated’ is not correct term in this context. Please replace it with a more suitable one e.g. characteristic.
19. L62-63: I guess you wanted to say that PSGs are the most abundant proteins of fetal origin or trophoblast origin in maternal serum during pregnancy not the most abundant of all maternal serum proteins during the late pregnancy. Please correct the sentence to correspond to the facts and add suitable reference.
20. L63-64: How is this relevant for results in this manuscript? Please explain the connection or the context in which you are writing these facts.
21. L67-69: How is this relevant for results in this manuscript? Please explain the connection or the context in which you are writing these facts. Why is this sentence in the middle of explanation of the connections of PSG1 and pregnancy pathologies?
22. L70-71: It is more suitable to write “...that PSG1 may be involved in EOPE pathophysiology” instead of “...PSG1 may regulate the development and incidence of EOPE”.
23. L88: Please write “may be involved in the pathophysiology of EOPE” instead of “may involve the pathophysiology of EOPE.”
24. L102: Please check and correct how many pregnant women you had drawn blood from for the control group. Here is written 4 and in the results of the revised manuscript it is 10 age-matched controls.
25. L126: Please change” when the cells reached to 60%-80% they were digested” to “when the cells reached 60%-80% confluency they were digested”.
26. L131: Please change HTR8 to HTR-8 throughout the manuscript.
27. L140-141: The sentence: “HTR8/SVneo cells were inoculated with a 60% density into a 96-well plate with a total volume of 100 μL cells per well of medium and suspension” is not understandable. Please rephrase it to give accurate and relevant information in an understandable way e.g. number of cells seeded per well, in complete or serum free medium, etc.
28. L150-152: You wrote that cells were placed in a buffer solution without Ca2+ and then you wrote that buffer consisted of 2 mM CaCl2 and other salts and compounds. Please rephrase this sentence to correspond the facts. Either there was or there wasn’t Ca2+ in the buffer. Please carefully check all the experimental procedures.
29. L161: What does ‘The experiment used three biological repeat sequences” mean? Did you mean three biological repeats? Please clarify.
30. L168: “RNA was ... reversed and transcribed” is not adequate terminology. I believe that you referred to the process of reversed transcription using reverse transcriptase. Please write this sentence in an adequate way.
31. L191: Please write “proteins were transferred” instead of “protein samples were placed”.
32. L192-193: Membranes were blocked with the 5% non-fat milk not the antibodies. Please rephrase this sentence in an adequate way.
33. L219-222: Figure 1 legend: “patients with the early-onset preeclampsia” instead “patients in the early-onset preeclampsia”.
34. L236-243: Figure 2 legend: ****P is missing and *P should be deleted.
35. L250: Please add full term before DEGs abbreviation.
36. L271: Please write “Ca2+ imaging experiments” instead of “Ca2+ image experiments”.
37. L323: Figure 6 legend: Please give adequate title for the Figure 6 to correspond presented results e.g. Akt signalling.
38. L323-333: Figure 6 legend: You can summarize all the P values appearing in the Figure 6 at the end of the legend.
39. L346-347: Please write “that impaired maternal spiral artery remodeling is a pathological feature of preeclampsia” instead of “that maternal spiral artery remodeling is a pathological feature of preeclampsia”.
40. L357-358: It is not clear what you wanted to say with “As a result”.
41. L362: Please write “The expression level of Orai1 protein and SOCE activity” instead of “The expression level and SOCE activity of the Orai1 channel protein”
42. L372-373: Please write “Essential mechanism for maintaining intracellular Ca2+ homeostasis and Ca2+ signalling regulation is SOCE.” instead of “One essential mechanism for controlling Ca2+ is SOCE.”
43. L378: Is it “also increasing SOCE” or SOCE increase is a consequence of upregulation of Orai1 protein expression?
44. L384: “numerous pieces of research” is inadequate terminology. Please change.
45. L391-393: Please add reference.
46. L397: Please rephrase this part of the sentence: “which may contribute to the development of EOPE.” You should explain how decreased serum PSG1 levels in EOPE patients could affect trophoblast function. It could be short explanation, in one sentence.
47. L399: Please write “This work offers PSG1 as a prospective target for treatment of EOPE” instead of “This work offers a prospective target of PSG1 to prevent and treat EOPE”.
48. L399-400: Please rephrase this part of the sentence, it is not clearly written: “and a plausible mechanism for its incidence and progression.” I guess by “its incidence” you meant “EOPE development”. Please clarify what you wanted to say.
Comments on the Quality of English LanguageComments on the quality of English and writing can be found in general comments on the revised manuscript.
Author Response
Point-by-point Response
General response: Thank you very much for your comments and professional advice. We have carefully evaluated the reviewer’ critical comments and thoughtful suggestions. These advices help us to improve the manuscript. All changes in the manuscript are marked in red. We hope that this revised manuscript has addressed all questions raised by the #1 reviewer.
Reviewers' Comments to Author:
- How come gestational age at blood sampling for EOPE group was 33.21±1.97 (Table 1)? Does it mean that at least one of the pregnant women in this group was more than 34 weeks pregnant at the time of blood sampling? If that so, how come you enrolled that/those women into EOPE group since EOPE is defined by new-onset hypertension and proteinuria before 34th week of gestation? Please provide results for women diagnosed with EOPE and if possible, for more than four EOPE patients.
Answer: Thanks for your questions. I am so sorry for our mistake to write a pregnant woman with a gestational age of 33 weeks and 5 days into 35 weeks and 3 days during data input. This has been corrected in this revised manuscript. Based on your suggestions, we have expanded the EOPE group data to 7 women and provided the original medical records in the attached file (diagnostic criteria highlighted), proving that all 7 women meet the EOPE diagnosis. Please check them again.
- Serum PSG1 levels in healthy pregnant women was much higher in your measurements, (256.53±18.95 μg/mL) compared to other published data e.g. (1080/01443615.2019.1679734, 10.5935/1518-0557.20210068, 10.1081/prg-120016794, https://doi.org/10.3390/cells8111369) where PSG1 serum levels were at tens of μg/mL or even in ng/ml although healthy pregnant woman participants were in the similar gestational age at the time of blood sampling. Please comment on that.
Answer: Thank you very much for your question. It is yes that the PSG1 concentration was reported in different range. The serum PSG1 concentration measured in the present study is similar to the results from other groups [DOI:10.11723/mtgyyx 1007-9564 202302009, DOI:10.1111/j.1471-0528.1979.tb10717.x ], but some other studies also reported that the serum concentration of PSG1 is much lower than our data [DOI:10.1080/01443615.2019.1679734,DOI:10.5935/1518-0557.20210068, DOI:10.1081/prg-120016794, https://doi.org/10.3390/cells8111369]. We speculate that this difference may be due to differences in detection methods and reagent kits. During our ELISA experiment, we have set up three repeated wells and operated according to the instructions of the ELISA kit to ensure the authenticity and validity of the experimental data. We have also explained this issue in the Discussion section to remind readers. Please refer to the page 12 and line 354-357.
- You chose 0.1 and 0.5 μg/mL PSG1 for treatment of HTR-8/SVneo cells in your experiments. Can you explain on which bases you chose these concentrations? That should also be written in the manuscript. In your answer on the same question after first revision you said that you conducted cell viability test with those PSG1 concentrations and that you “found that 0.5 μg/mL PSG1 was enough to induce cell viability change and 0.5 μg/mL concentration was chosen for our experiment.” First, I couldn’t find the graph of pre-experiments you are referring to in your answer after first revision. Second, cell viability assay results from the manuscript (Figure 2C) show that already 0.1 μg/mL PSG1 increased HTR-8/SVneo cell proliferation significantly.
Answer: We were really sorry for our mistakes. We have made an mistake of PSG1 concentration in the first-round reply. What we want to say is as follows: “Cell viability assays were carried out with three pretreated PSG1 concentrations: 0, 0.1 and 0.5 μg/mL. It was found that 0.1 μg/mL PSG1 was enough to induce cell viability change, and 0.1 μg/mL concentration was chosen for our experiment. The results are displayed in the following figure:
- Next, why did you choose PSG1 concentrations that stimulate cell proliferation for wound healing assay instead of lower concentration which do not change cell proliferation? In this way, cell proliferation stimulation could interfere with the cell migration results. Looking at your results (Figure 2B and 2C), rates of PSG1 stimulation of HTR-8/SVneo cell migration are almost the same as cell proliferation stimulation rates for the same PSG1 concentrations. Also, in your answers after first revision you said: “In order to minimize the impact of cell proliferation on the migration experiments, the scratched cells were cultured in a serum-free basic culture medium.” Did you perform cell viability assays in the same conditions (0.1 and 0.5 μg/mL PSG1, serum-free basic medium, 24 h treatment)? If yes present results and comment on them. If not, I recommend checking if PSG1 affects cell proliferation in serum-free conditions.
Answer: Thank you very much for your valuable suggestion. We have performed cell viability assays in serum-free conditions as your suggestion. It was found that PSG1 did not significantly promote HTR-8/SVneo cell proliferation in serum-free conditions. The results are displayed in the following figure. Therefore, we believe that the cell proliferation does not affect our result of cell migration. We added this data into Figure 2. Please check again.
- Please discuss migration assay results in connection with cell proliferation results in manuscript also.
Answer: Thank you very much for your suggestion. We added new data and an explanation into this revised version.
- Please add why you chose 0.1 μg/ml PSG1 treatment for transcriptome analysis in connection with the other results in the manuscript.
Answer: Thank you very much for your valuable suggestion. We have added the sentence to explain why we choose the concentration as follows: “Based on the above results, it was found that 0.1 μg/mL PSG1 was enough to induce the changes in cell viability and migration. Therefore, we choose 0.1 μg/mL concentration to treat HTR-8/SVneo cells and then performed RNA-seq experiment.” Please check again.
- Please write in more details how you treated cells for Ca2+ imaging experiments before adding Fluo-8 AM in Material and Methods section (2.6). How long was treatment with PSG1 and MRS1845?
Answer: Thank you very much for your valuable suggestion. Following your suggestion, we have revised this sentence as follows: “In some experiments, the cells were treated with PSG1 (0.1 µg/mL) and/or MRS1845 (10 µM) for 24 h”
- Please add experimental procedures for treating cells with MRS1845 and ZSTK474 in Wound healing experiments and for Western blot experiments.
Answer: Thank you very much for your valuable suggestion. Following your suggestions, we have added or revised some sentences as follows:
In Western blot experiments, we added the sentences as “HTR-8/SVneo cells were collected and lysed using RIPA buffer (#P0013B; Beyotime Biotech, Jiangsu, China) to extract total proteins. In some experiments, the cells were treated with MRS1845 or ZSTK474 with or without PSG1 for 24 h.”
In Wound healing experiments, we have revised this sentence as: “ Next, the original culture medium was discarded, and the cells were cultured in a serum-free basic culture medium. In some experiments, the cells were treated with MRS1845 or ZSTK474 with or without PSG1 for 24 h.”
- Please give a little bit more information in the Introduction section about EOPE and LOPE, the differences between these two PE subtypes regarding clinical manifestations and suggested pathophysiology. Discuss shortly why you focused on EOPE.
Answer: Thank you very much for your suggestion. The following are the differences between EOPE and LOPE and reasons for focusing on EOPE. We added some information into the Introduction section and please check again.
The differences between EOPE and LOPE regarding clinical manifestations: Preeclampsia is stratified depending on the time of onset: (1) early-onset preeclampsia (EOPE) is occurred before 34 weeks of gestation, and (2) late-onset preeclampsia (LOPE) is occurred from 34 weeks of gestation. Although EOPE and LOPE share the similar clinical symptoms, these two subtypes of preeclampsia lead to different outcomes. EOPE, although less common, is associated with higher rates of neonatal mortality and a greater degree of maternal morbidity compared to LOPE. Therefore, EOPE has received more attention from clinical researchers. (Reference: Chen H, Aneman I, Nikolic V, Karadzov Orlic N, Mikovic Z, Stefanovic M, Cakic Z, Jovanovic H, Town SEL, Padula MP, McClements L. Maternal plasma proteome profiling of biomarkers and pathogenic mechanisms of early-onset and late-onset preeclampsia. Sci Rep. 2022 Nov 9;12(1):19099. doi: 10.1038/s41598-022-20658-x.)
The differences between EOPE and LOPE regarding suggested pathophysiology: Study has shown that impaired placental development in early pregnancy and subsequent growth restriction is often associated with EOPE, while LOPE is likely associated with maternal endothelial dysfunction. (Reference: Marín R, Chiarello DI, Abad C, Rojas D, Toledo F, Sobrevia L. Oxidative stress and mitochondrial dysfunction in early-onset and late-onset preeclampsia. Biochim Biophys Acta Mol Basis Dis. 2020;1866(12):165961. doi:10.1016/j.bbadis.2020.165961)
We previously found a decrease in the serum level of PSG1 in patients with EOPE [Reference]. In addition, abnormal placental development is an important pathological mechanism of EOPE. This finding raises the possibility that PSG1 may affect the migration of trophoblast cells which is related to placental development in EOPE pathophysiology.
(Reference: Tu C, Tao F, Qin Y, et al. Serum proteins differentially expressed in early- and late-onset preeclampsia assessed using iTRAQ proteomics and bioinformatics analyses. PeerJ. 2020;8:e9753. Published 2020 Sep 1. DOI:10.7717/peerj.9753.)
- L42-72: I would recommend reorganizing this section in a way that in the first paragraph you present relevant data about preeclampsia as you started it and in the second paragraph to give relevant information about PSGs and PSG1. Regarding that, I would recommend moving sentence L46-47 to the second paragraph.
Answer: Thank you very much for your suggestion. We have moved the sentence to the second paragraph.
- L30: Please add the word “trophoblast” before “proliferation and migration”.
Answer: Thank you very much for your careful check. We have added the word as your suggestion.
- L34-36: Please rephrase this sentence. The part: “MRS1845...also reducing the promoting effect of PSG1 via the Akt signaling pathway” is not clear.
Answer: Thanks for your careful check. We have rewritten the sentence as following: “The selective inhibitor of Orai1 (MRS1845) weakened the migration promoting effect mediated by PSG1 via suppressing the Akt signaling pathway.”
- L44-45: Please add the word “new-onset” before “hypertension and proteinuria” and add suitable reference for this statement.
Answer: Thank you very much for your careful check. We have added “new-onset” before “hypertension and proteinuria” and a suitable reference.
Reference1: Chaiworapongsa T, Chaemsaithong P, Yeo L, Romero R. Pre-eclampsia part 1: current understanding of its pathophysiology. Nat Rev Nephrol. 2014 Aug;10(8):466-80. DOI: 10.1038/nrneph.2014.102. Epub 2014 Jul 8.
- L47: Please add the word “maternal” before “bloodstream.”
Answer: Thank you very much for your careful check. We have added the word as your suggestion.
- L54: Please use PE and other abbreviations throughout the manuscript after introducing them.
Answer: Thank you very much for your careful check. We have corrected errors as your suggestion.
- L54-58: Sentence is still not precise and clear enough. Please reorganize it this way: In PE, the migration and invasion abilities of trophoblast cells are reduced, and remodeling of the uterine spiral arteries is impaired, leading to placental ischemia and hypoxia, causing a series of oxidative stress reactions, thereby causing a series of several symptoms of PE.
Answer: Thank you very much for your careful check. We have rewritten the sentence as your suggestion.
- L58: Please write “mechanisms” instead of “mechanism”.
Answer: Thank you very much for your careful check. We have corrected the error as your suggestion.
- L61: “Dedicated’ is not correct term in this context. Please replace it with a more suitable one e.g. characteristic.
Answer: Thank you very much for your careful check. We have replaced " Dedicated " with " characteristic " in this revised manuscript.
- L62-63: I guess you wanted to say that PSGs are the most abundant proteins of fetal origin or trophoblast origin in maternal serum during pregnancy not the most abundant of all maternal serum proteins during the late pregnancy. Please correct the sentence to correspond to the facts and add suitable reference.
Answer: Thank you very much for your valuable suggestion. We have corrected the sentence as follows: “PSGs are the most abundant trophoblastic origin proteins in maternal blood during human pregnancy” and added a suitable reference.
Reference 9: Moore T, Dveksler GS. Pregnancy-specific glycoproteins: complex gene families regulating maternal-fetal interactions. Int J Dev Biol. 2014;58(2-4):273-280. DOI:10.1387/ijdb.130329gd.
- L63-64: How is this relevant for results in this manuscript? Please explain the connection or the context in which you are writing these facts.
Answer: Thank you very much for your criticism. After our check, there is no significant correlation between these contents and the results. We have deleted the sentence in this revised manuscript.
- L67-69: How is this relevant for results in this manuscript? Please explain the connection or the context in which you are writing these facts. Why is this sentence in the middle of explanation of the connections of PSG1 and pregnancy pathologies?
Answer: Thank you very much for your criticism. After our check, there is no significant correlation between these contents and the results. We have deleted the sentence in this revised manuscript.
- L70-71: It is more suitable to write “...that PSG1 may be involved in EOPE pathophysiology” instead of “...PSG1 may regulate the development and incidence of EOPE”.
Answer: Thank you very much for your careful check. We have replaced “PSG1 may regulate the development and incidence of EOPE” with “PSG1 may be involved in EOPE pathophysiology”.
- L88: Please write “may be involved in the pathophysiology of EOPE” instead of “may involve the pathophysiology of EOPE.”
Answer: Thank you very much for your careful check. We have replaced “may be involved in the pathophysiology of EOPE” with “may involve the pathophysiology of EOPE”.
- 24. L102: Please check and correct how many pregnant women you had drawn blood from for the control group. Here is written 4 and in the results of the revised manuscript it is 10 age-matched controls.
Answer: Sorry for the writing problems. We have corrected it into "ten" in this revised manuscript.
- L126: Please change” when the cells reached to 60%-80% they were digested” to “when the cells reached 60%-80% confluency they were digested”.
Answer: Thank you very much for your careful check. We have revised the sentence as your suggestion in this revised manuscript.
- L131: Please change HTR8 to HTR-8 throughout the manuscript.
Answer: Sorry for the writing problems. We have corrected it into "HTR-8" throughout the manuscript.
- L140-141: The sentence: “HTR8/SVneo cells were inoculated with a 60% density into a 96-well plate with a total volume of 100 μL cells per well of medium and suspension” is not understandable. Please rephrase it to give accurate and relevant information in an understandable way e.g. number of cells seeded per well, in complete or serum free medium, etc.
Answer: Thank you very much for your careful check. We have rewritten the sentence as follows: “HTR-8/SVneo cells were cultured in a 96-well plate at 5000 cells/well with RIPM1640 medium supplying 10% fetal bovine serum.”
- L150-152: You wrote that cells were placed in a buffer solution without Ca2+ and then you wrote that buffer consisted of 2 mM CaCl2 and other salts and compounds. Please rephrase this sentence to correspond the facts. Either there was or there wasn’t Ca2+ in the buffer. Please carefully check all the experimental procedures.
Answer: We were really sorry for our mistakes. We have corrected it as follows: “After then, the cells were placed in a buffer solution without Ca2+ (OPSS, 140 mM NaCl, 5 mM KCl, 1 mM MgCl2, 10 mM glucose, and 5 mM HEPES, pH 7.3 to 7.4 adjusted with NaOH).” in this revised manuscript.
- L161: What does ‘The experiment used three biological repeat sequences” mean? Did you mean three biological repeats? Please clarify.
Answer: Sorry for the writing problems. We have corrected it into "three biological repeats".
- L168: “RNA was ... reversed and transcribed” is not adequate terminology. I believe that you referred to the process of reversed transcription using reverse transcriptase. Please write this sentence in an adequate way.
Answer: Thank you very much for your careful check. We have rewritten the sentence to make it specific as follows: “And then the purified mRNA was sheared using the fragmentation buffer. The fragmented mRNAs were then used for the first-strand cDNA synthesis using reverse transcriptase and random hexamer primers. The second-strand cDNA was synthesized using DNA polymerase I and RNase H. After the fragments were ligated to adaptors, they were isolated as polymerase chain reaction (PCR) templates.” in this revised manuscript.
- L191: Please write “proteins were transferred” instead of “protein samples were placed”.
Answer: Thank you very much for your careful check. We have replaced “protein samples were placed” with “proteins were transferred” in this revised manuscript.
- L192-193: Membranes were blocked with the 5% non-fat milk not the antibodies. Please rephrase this sentence in an adequate way.
Answer: Thank you very much for your careful check. We have revised the sentence as follows: “After being blocked in 5% non-fat milk for 2 h, the membranes were incubated overnight at 4°C with primary antibodies.”.
- L219-222: Figure 1 legend: “patients with the early-onset preeclampsia” instead “patients in the early-onset preeclampsia”.
Answer: Thank you very much for your careful check. We have replaced " patients in the early-onset preeclampsia” with “patients with the early-onset preeclampsia”.
- L236-243: Figure 2 legend: ****P is missing and *Pshould be deleted.
Answer: Thank you very much for your careful check. We have added “****P < 0.0001” and deleted “*P <0.05” in this revised manuscript.
- L250: Please add full term before DEGs abbreviation.
Answer: Thank you very much for your careful check. We have added “differentially expressed genes” before DEGs abbreviation where it was first written in the Method section.
- L271: Please write “Ca2+ imaging experiments” instead of “Ca2+ image experiments”.
Answer: Thank you very much for your careful check. We have replaced " Ca2+ image experiments” with “Ca2+ imaging experiments”.
- L323: Figure 6 legend: Please give adequate title for the Figure 6 to correspond presented results e.g. Akt signalling.
Answer: Thank you very much for your careful check. We have revised the title as follows : “Effect of Pregnancy-Specific Glycoprotein 1 (PSG1) on migration of HTR-8/SVneo cells through the Orai1/Akt signaling pathway.”
- L323-333: Figure 6 legend: You can summarize all the P values appearing in the Figure 6 at the end of the legend.
Answer: Thank you very much for your careful check. We have summarized all the P values according to your suggestion. Please check it.
- L346-347: Please write “that impaired maternal spiral artery remodeling is a pathological feature of preeclampsia” instead of “that maternal spiral artery remodeling is a pathological feature of preeclampsia”.
Answer: Thank you very much for your careful check. We have added the word as your suggestion in this revised manuscript.
- L357-358: It is not clear what you wanted to say with “As a result”.
Answer: Thank you very much for your careful check. We have replaced "As a result” with “For this reason”.
- L362: Please write “The expression level of Orai1 protein and SOCE activity” instead of “The expression level and SOCE activity of the Orai1 channel protein”
Answer: Thanks for your careful check. We have replaced " One essential mechanism for controlling Ca2+ is SOCE. " with " Essential mechanism for maintaining intracellular Ca2+ homeostasis and Ca2+ signaling regulation is SOCE. ". Please check it.
- L372-373: Please write “Essential mechanism for maintaining intracellular Ca2+ homeostasis and Ca2+ signalling regulation is SOCE.” instead of “One essential mechanism for controlling Ca2+ is SOCE.”
Answer: Thanks for your careful check. We have replaced " One essential mechanism for controlling Ca2+ is SOCE. " with " Essential mechanism for maintaining intracellular Ca2+ homeostasis and Ca2+ signaling regulation is SOCE. ".
- L378: Is it “also increasing SOCE” or SOCE increase is a consequence of upregulation of Orai1 protein expression?
Answer: Thanks for your careful check. We have added a sentence as follows: “Meanwhile, SOCE in HTR-8/SVneo cells also increased with the increment of Orai1 protein expression.” in the revised manuscript.
- L384: “numerous pieces of research” is inadequate terminology. Please change.
Answer: Thanks for your careful check. We have replaced " numerous pieces of research " with " much research ".
- L391-393: Please add reference.
Answer: Thank you very much for your careful check. We have added the reference in the revised manuscript.
(Reference 41: Radoslavova S, Folcher A, Lefebvre T, et al. Orai1 Channel Regulates Human-Activated Pancreatic Stellate Cell Prolif-eration and TGFβ1 Secretion through the AKT Signaling Pathway. Cancers (Basel). 2021;13(10):2395. Published 2021 May 15. DOI:10.3390/cancers13102395.)
- L397: Please rephrase this part of the sentence: “which may contribute to the development of EOPE.” You should explain how decreased serum PSG1 levels in EOPE patients could affect trophoblast function. It could be short explanation, in one sentence.
Answer: Thanks for your careful check. We have rephrased the sentence as follows: “Additionally, PSG1 stimulates migration in HTR-8/SVneo cells via the Orai1/Akt pathway, thereby reduced PSG1 secretion may cause insufficient remodeling of spiral arteries and inhibit the occurrence and development of EOPE.”
- L399: Please write “This work offers PSG1 as a prospective target for treatment of EOPE” instead of “This work offers a prospective target of PSG1 to prevent and treat EOPE”.
Answer: Thanks for your careful check. We have rewritten the sentence according to your suggestion as follows: “This work offers a prospective target of PSG1 to prevent and treat EOPE.”
- L399-400: Please rephrase this part of the sentence, it is not clearly written: “and a plausible mechanism for its incidence and progression.” I guess by “its incidence” you meant “EOPE development”. Please clarify what you wanted to say.
Answer: Thank you very much for your constructive suggestion. We have replaced " incidence " with " development " in this revised manuscript. Please check it.

Reviewer 2 Report
Comments and Suggestions for Authors
manuscript has been significantly improved after revision and can be accepted in the present form
Author Response
Thank you very much for your positive comments.
Reviewer 3 Report
Comments and Suggestions for Authors
Dear Authors,
I accept your reply
Author Response

(The authors gave the same response as above.)

Round 3
Reviewer 1 Report
Comments and Suggestions for Authors
The authors answered the questions, accepted remarks and significantly improved the manuscript. I have 2 substantial corrections to ask the authors for to introduce in the manuscript and a few smaller writing corrections before the manuscript is published.
L409-410: Please correct this part of the sentence “thereby reduced PSG1 secretion may cause insufficient remodeling of spiral arteries and inhibit the occurrence and development of EOPE.” Please add: “may cause insufficient trophoblast migration and remodeling of spiral arteries” to the sentence. Also, It should be written: “and induce the occurrence and development EOPE” not inhibit.
L410-411: Please replace this sentence with: “This work offers PSG1 as a prospective target for treatment of EOPE.”
Comments on the Quality of English LanguageL86: Please change “extra villous trophoblast” to “extravillous trophoblast”.
L122: Please change “Professor Graham of Canada” to “Professor Charles H Graham from Queen's University, Kingston, ON, Canada”.
L122-123: Please change “It was cultivated” to “Cells were cultivated”.
L203: Please change "to identify the protein” to “to visualize protein bands”.
L227: “In the cell wound healing experiment”, you can erase “cell” from this part of the sentence.
L245-248: Please change ”showing the survival of HTR-8/SVneo cells to ”showing the proliferation of HTR-8/SVneo cells”.
L289-290: Please correct “The loading control was the GAPDH protein” in Figure 4 legend. In Figure 4 A and B β-Tubulin was shown.
L299: Please add “ns, non-significant”.
Author Response
Point-by-point Response
General response: Thank you very much for your comments and professional advice. We have carefully revised the manuscript again. Please check everything again.
Comments and Suggestions for Authors
The authors answered the questions, accepted remarks and significantly improved the manuscript. I have 2 substantial corrections to ask the authors for to introduce in the manuscript and a few smaller writing corrections before the manuscript is published.
L409-410: Please correct this part of the sentence “thereby reduced PSG1 secretion may cause insufficient remodeling of spiral arteries and inhibit the occurrence and development of EOPE.” Please add: “may cause insufficient trophoblast migration and remodeling of spiral arteries” to the sentence. Also, it should be written: “and induce the occurrence and development EOPE” not inhibit.
Answer: Thank you very much for your valuable suggestion. We have corrected the sentence as follows: “thereby reduced PSG1 secretion may cause insufficient trophoblast migration and remodeling of spiral arteries and induce the occurrence and development of EOPE.”
L410-411: Please replace this sentence with: “This work offers PSG1 as a prospective target for treatment of EOPE.”
Answer: Thank you very much for your suggestion. We have replaced the sentence with: “This work offers PSG1 as a prospective target for treatment of EOPE.”
Comments on the Quality of English Language
L86: Please change “extra villous trophoblast” to “extravillous trophoblast”.
Answer: Thanks for your suggestion. We have changed “extra villous trophoblast” to “extravillous trophoblast”.
L122: Please change “Professor Graham of Canada” to “Professor Charles H Graham from Queen's University, Kingston, ON, Canada”.
Answer: Thanks for your suggestion. We have changed “Professor Graham of Canada” to “Professor Charles H Graham from Queen's University, Kingston, ON, Canada”.
L122-123: Please change “It was cultivated” to “Cells were cultivated”.
Answer: Thanks for your criticism. We have changed “It was cultivated” to “Cells were cultivated”.
L203: Please change "to identify the protein” to “to visualize protein bands”.
Answer: Thanks for your suggestion. We have replaced "to identify the protein” with “to visualize protein bands”.
L227: “In the cell wound healing experiment”, you can erase “cell” from this part of the sentence.
Answer: Thanks for your suggestion. We have deleted “cell” from the sentence.
L245-248: Please change ”showing the survival of HTR-8/SVneo cells to ”showing the proliferation of HTR-8/SVneo cells”.
Answer: Thanks for your criticism. We have changed “survival” to “proliferation”.
L289-290: Please correct “The loading control was the GAPDH protein” in Figure 4 legend. In Figure 4 A and B β-Tubulin was shown.
Answer: So sorry for our writing error. We have corrected “GAPDH” to “β-Tubulin”.
L299: Please add “ns, non-significant”.
Answer: Thanks for your suggestion. We have added “ns, non-significant” at the end of the sentence.